# A Multilayer Perception for Estimating the Overall Risk of Residential Projects in the Conceptual Stage

**Mohamed Badawy** [1], **Fahad K. Alqahtani** [2,*] **and Mohamed Sherif** [3]

1    Structural Engineering Department, Ain Shams University, Cairo 11517, Egypt;
     mohamed_badawy@eng.asu.edu.eg
2    Civil Engineering Department, College of Engineering, King Saud University, P.O. Box 800,
     Riyadh 11421, Saudi Arabia
3    Department of Civil and Environmental Engineering, College of Engineering, The University of Hawai'i at
     Mānoa, 2540 Dole Street, Honolulu, HI 96822, USA; msherif@hawaii.edu
*    Correspondence: bfahad@ksu.edu.sa

**Abstract:** The ability to foresee hazards early plays a critical role in estimating the entire cost of a project. Although several studies have established models to predict the total cost of a project at a conceptual stage, there remains a research vacuum in measuring the overall risk at this stage. Using artificial neural networks, this research provides a strategy for estimating the overall risk in residential projects at the conceptual stage. There are eight important components in the suggested paradigm. The model was created using data from 149 projects. In the first hidden layer in the model, there are five neurons, and in the second hidden layer, there are three neurons. The suggested model's mean absolute error rate was 11.7%. In the conceptual stage of residential projects, the number of floors, the type of interior finishes, and the implementation of risk management processes are the significant aspects that influence the overall risk. The proposed model assists project managers in precisely estimating the project's overall risk, which leads to a more accurate estimation of the contract's entire worth at the conceptual stage, allowing the stakeholders to decide whether or not to proceed with the project.

**Keywords:** early-stage; overall risk; residential projects; a multilayer perception

## 1. Introduction

A project's cost should be projected with a high degree of precision; however, making a conceptual cost estimate is challenging at this time due to a lack of data [1]. Construction companies require an early budget estimate to assess whether this expenditure is acceptable and, hence, whether the project should be continued or abandoned. To estimate the contract value, the project manager usually calculates the direct cost, indirect cost, profit, and contingencies for project risks. As a result, performing an early risk assessment is critical. Stakeholders can make decisions and choices during the conceptual stage of the projects, which have major impacts on construction duration and costs, but this effect reduces as the project progresses through its life cycle [2]. Negative risks may result in schedule delays and expense overruns [3]. As a result, the project manager should concentrate as much as possible on the major risks [4]. Overall risk estimation suffers many challenges during the conceptual stage of a project due to the limited data provided. A major issue that develops in the early phase is the lack of effective and reliable overall risk estimation approaches. As a result, project decision-makers have begun to focus more on conceptual planning, where a thorough cost analysis is a critical component in achieving the project's objectives [5]. The goal of the estimation process is to make sure that the contract plans and specifications match the cost of completing the project [6].

Developing cost estimation models, both in the planning stage and in the conceptual stages of a project, has been the subject of a lot of research. Despite the importance of overall

project risk, there is a gap in the research on assessing the overall risk in the conceptual stage of a project when there is insufficient project information, necessitating the development of a model for predicting the overall risk at the conceptual stage of a project [7]. This research uses artificial neural networks to construct a model for evaluating the overall risk in residential projects, based on a few characteristics that may be easily recognized at a conceptual stage with acceptable accuracy. In other words, this research does not investigate the assessment of individual positive or negative risk variables, but rather the classification of the overall risk of residential projects at a conceptual stage based on the influence on project cost.

As there is a gap in developing a method to estimate the total risk in the conceptual stage, this research aims to propose a model to predict the overall risk of residential buildings at the conceptual stage using an artificial neural network with a multilayer perception.

## 2. Literature Review

Financial risks are regarded as the most significant risks in construction projects in Egypt and Saudi Arabia, followed by design, political, and construction risks [8]. Lack of money, a tight deadline, design revisions, insufficient information on sustainable design, and a weak definition of sustainable scope were the top hazards to sustainable building projects in the UAE [9]. At the planning stage, risk assessment models have been proposed, using an artificial neural network such as risk assessment in Saudi Arabian building projects [10] or the construction of an expressway [11]. Another model, based on system dynamics and discrete event simulation, was proposed to evaluate the impact of risk factors on project schedules in infrastructure projects [12]. Al-Tabtabai and Alex (2000) provided an ANN-based model for predicting project cost escalation owing to political concerns and the average error was 7% [13]. A model using the Bayesian Belief Network was developed to assess and enhance the implementation of residential construction projects. Improper construction procedures and poor communication were the top risk factors [14]. At a conceptual stage, the most important aspects that determine the overall risk are the use of risk management processes, the entire project duration, contract cost, and contract type [15]. The most essential criteria in the tender process are price, the scope of work, and technical resources [16].

At the conceptual stages, there is a research gap in estimating overall risk. As a result, an essential point to consider is what proportions of errors are acceptable in any model assessing the overall risk. For every equation or model, there is a rate of error, but how can this ratio be judged, meaning how one can determine if the model is accepted or not. It is not fair to judge the error rate of a model in the conceptual stage, where there is not enough information, to the error rate of a model in the design stage where there is sufficient information. It is expected that the error rate is less in the design stage than at the conceptual stage. Therefore, the error rate of any model must be compared with the extent of errors in the same stage. Since there is no research that deal with estimating the total risk of the project in the conceptual stage, the error rate in the proposed model for calculating the total risk of the project in the conceptual stage was compared with the acceptable range of the rates of estimation models for cost estimation models in the conceptual phase. Hence, to estimate the allowable range of percentage errors, the authors relied on a review of past studies on cost estimation at a conceptual stage.

To estimate the cost of school buildings in Korea, ten factors were identified. Three models were developed to calculate the cost of the school buildings, based on 217 projects. The first model was developed using neural network techniques, while the regression analysis was used in the second model and the third model was presented using the support vector machine. The results of the neural network model showed a more accurate estimate than the results of regression analysis or the supporting vector machine models [17]. Two studies were conducted in Gaza to estimate the cost of buildings at an early stage. The first research was based on seven variables and a model was proposed based on information derived from 71 construction projects using artificial neural networks [18]. While the

second research developed a model for assessing the cost of construction projects with a high degree of accuracy and without the need for a lot of information, through the use of artificial neural networks. A database of 169 projects was collected from relevant institutions in the Gaza Strip has been adopted. The artificial neural network model has eleven factors as independent inputs [19]. A study to predict the cost of construction projects at the conceptual stage in Taiwan using ten parameters. The research suggested the utilization of the evolutionary fuzzy neural inference model to enhance cost assessment accuracy. The proposed model was relied on eleven factors [20]. In Egypt, a model to assess the cost of a residential building at an early stage using the artificial neural network and data obtained from 174 residential projects. The proposed model depended on four parameters: number of floors, the area of the floor, type of external finishing, and type of internal finishing [21]. The costs of 136 executed projects were utilized to propose an artificial neural network model to predict the preliminary cost of construction projects in Yemen. The suggested model contained 17 factors [22]. In the United States, research was conducted on the difference in the computation of construction costs utilizing artificial neural networks by comparing nineteen variables in 20 projects [23]. In Taiwan, a study has presented a prototype for the rapid assessment of a proposal integrating a probabilistic cost sub-model and a multi-factor assessment sub-model. The cost-based sub-model concentrates on the cost divisions. While the multi-factor assessment sub-model captures the specific elements influencing the cost division. That research is based on 21 variables [24]. The eight previous studies mostly agreed on nine primary factors that can be used for cost estimates at the conceptual stage of a project. These nine parameters are floor area, number of floors, type of foundation, number of elevators, type of slab, type of exterior finishing, interior finishes, type of electromechanical works, and number of basements. Table 1 shows the different sources for each parameter.

**Table 1.** Sources of parameters influencing the cost estimation.

| Factor | [17] | [18] | [19] | [20] | [21] | [22] | [23] | [24] |
|---|---|---|---|---|---|---|---|---|
| Floor area | √ | √ | √ | √ | √ | √ | √ | √ |
| Number of floors | √ | √ | √ | √ | √ | √ | √ | √ |
| Slab type | √ | | √ | | | √ | √ | √ |
| Internal finishes | √ | | √ | √ | √ | √ | √ | √ |
| Number of elevators | | √ | √ | | | √ | √ | √ |
| External finishes | | | √ | | √ | √ | √ | √ |
| Foundation type | | √ | √ | | | √ | √ | √ |
| Basement | √ | | √ | | | √ | √ | |
| Electromechanical type | | | √ | √ | | √ | √ | √ |

The symbol "√" means the corresponding research determined the corresponding factor as a key factor for identifying the cost estimate at the conceptual stage.

Traditional cost estimation strategies in construction projects are the most often used. They rely on time-consuming manual project cost estimation or Excel spreadsheets, rather than using computerized tools to estimate building costs. Soft computing strategies for conceptual-stage software development were compared by Bhatnagar and Ghose (2012). The feed-forward back propagation neural network model had a mean absolute percentage error (MAPE) of 13%, a cascaded feed-forward back propagation neural network model had a MAPE of 13.6 percent, a layer recurrent neural network model had a MAPE of 11.5 percent, and a fuzzy logic model had a MAPE of 3.9 percent. This means they accepted models up to MAPE with a 13.5% acceptance rate [25].

There is a lot of research that investigates the cost estimates at the conceptual stage. Each research proposed a model with a mean absolute percentage error. Table 2 shows the mean absolute percentage error for some of these prior studies. Which illustrated that the errors in the proposed models were ranged from 4–28.2%. This means that the maximum acceptable mean percentage error in the proposed model at the conceptual stage is 28.2%.

**Table 2.** The minimum absolute percentage errors for the previous studies.

| Reference | Research | MAPE |
|:---:|:---:|:---:|
| [26] | Data modelling and the application of a neural network approach to the prediction of total construction costs | 16.6 |
| [27] | A neural network approach for early cost estimation of structural systems of buildings | 7 |
| [28] | Conceptual cost estimates using evolutionary fuzzy hybrid neural network for projects in the construction industry | 10.4 |
| [18] | Early-stage cost estimation of buildings construction projects using artificial neural networks | 4 |
| [19] | Cost estimation for building construction projects in the Gaza Strip using an artificial neural network (ANN) | 6 |
| [17] | Comparison of school buildings' construction costs' estimation methods using regression analysis, neural network, and support vector machine | 5.27 |
| [29] | Estimating water treatment plants' costs using factor analysis and artificial neural networks | 21.2 |
| [30] | Conceptual cost estimation model for engineering services in public construction projects | 28.2 |
| [31] | Cost estimation of civil construction projects using machine learning paradigm | 6.2 |
| [32] | Comparison of artificial intelligence techniques for project conceptual cost prediction | 26.3 |
| [21] | A hybrid approach for a cost estimate of residential buildings at the early stage | 13.2 |

Limited studies in the conceptual stage of risk estimation identified four criteria: use of risk management processes, duration of the entire project, total cost, and type of contract. As the total cost of the project can be estimated through nine criteria: floor area, number of floors, type of foundation, number of elevators, type of slab, type of external finishing, internal finishes, type of electromechanical works, and number of basements. Hence, the cost of the project can be replaced by these nine factors. The type of contract was not included due to research limitations, as the research is related to estimating the cost of housing projects based on a fixed price contract only. Hence, the initial list of criteria used to derive the overall project risk at the conceptual stage contained eleven criteria: floor area, number of floors, type of foundation, number of elevators, type of slab, type of exterior finish, interior finishes, type of electromechanical works, number of basements plus the use of risk management processes, and the entire project duration.

### 3. Methodology

This study's approach is divided into three sections. The first step in achieving the study's goal is to identify the essential components influencing the overall risk assessment by evaluating past research studies that focus on construction cost and risk estimation at a conceptual stage. As a result, eleven construction elements (or processes) were presented in the primary list. Five experts with at least 15 years of experience in the construction of residential projects were randomly selected. The primary questionnaire was presented to the experts using the Delphi technique to determine the parameters in the final questionnaire in three stages. In the first stage, experts were asked to add any missing criteria, if any, that could affect the overall risk and could be discovered in the conceptual stage. Data are collected from experts, revised, and re-sent back to the experts where they are asked to rank each criterion on a five-point Likert scale. After collecting the data from the second round, the averages are calculated and any factor that has a very low impact on estimating the overall risk of the project is removed from the final list. In the third round, the experts are asked to assess whether or not they agree with the final list. The third step is to develop the model. The model is simulated using artificial neural networks using Statistical Package for the Social Sciences (SPSS) software. The critical parameters that affect the estimation of

the overall risks at the conceptual stages are considered as inputs to the model while the output is the overall risk of the project. The model may contain one or two hidden layers. The number of neurons in the one-hidden-layer model can be three, four, or five. In models with two hidden layers, the number of neurons can be four in the first layer and three in the second, or five in the first layer and three in the second, or five in the first layer and four in the second. Thus, there are three different groups in terms of the number of neurons in each hidden layer. Hence, six models can be developed. The hyperbolic tangent function was used as an activation function for the hidden layers in six models, and the Sigmoid function was tested as an activation function for the hidden layers in six other models. Hence, twelve Multilayer Perceptron models have been identified and tested. To evaluate the performance of the model, the available data were also randomly divided 5-fold. The first fold contains 29 cases, while the subsequent folds contain 30 cases. Four folds were used to train the network in each model, while the fifth fold was used to evaluate the model. The final proposed model for estimating the overall risk in the conceptual stages is the model with the lowest mean absolute error rate.

## 4. Identifying the Critical Parameters Affecting the Estimation of Overall Risk at the Conceptual Stages

Literature analysis and earlier research yielded eleven criteria that can be utilized to forecast overall risk in residential projects. Floor space, number of floors, slab type, interior finishes, number of elevators, external finishes, electromechanical type, number of basements, foundation type, risk management implementation, and overall project duration are some of the elements to consider. Five experts with at least 15 years of experience in residential project management used the Delphi technique to select the final parameters. Table 3 shows the demographic information about the experts. The experts were requested to add missing parameters, if any, that could affect the overall risk and could be discovered at a conceptual stage, in the first round. There is no missing factor according to the experts' responses. On a five-point Likert scale, the experts were asked to evaluate the weight of each parameter in the second round. "1" indicates that this element is inconsequential; "2" suggests low importance; "3" indicates moderate significance; "4" indicates high significance; and "5" indicates extremely significant. Equation (1) was used to calculate the relative relevance index based on the responses received. Table 4 displays the relative importance indexes. The lowest number on the Likert scale is "1," and the highest is "5", resulting in a range of four, which will be graded according to the five categories. The zone for each category is 0.8. The very low category has a range of 1 to 1.8. The low category has a range from 1.8 to 2.6, while the range of the medium category is from 2.6 to 3.4. The high category is from 3.4 to 4.2, whereas the very high category is from 4.3 to 5.0.

$$RII = \frac{\sum W}{AN} = \frac{(5n_5 + 4n_4 + 3n_3 + 2n_2 + 1n_1)}{5N} \tag{1}$$

**Table 3.** Demographic data regarding the experts.

| Expert I.D. | Education Level | Experience | Job Title | Company |
|---|---|---|---|---|
| Expert (1) | Bachelor of Civil Engineering | 18 | Project manager | Private |
| Expert (2) | Bachelor of Civil Engineering | 15 | Risk manager | Private |
| Expert (3) | Ph.D. in Civil Engineering | 22 | Project manager | Private |
| Expert (4) | Bachelor of Civil Engineering | 17 | Project manager | Public |
| Expert (5) | Bachelor of Civil Engineering | 16 | Project manager | Private |

**Table 4.** The relative importance indices of critical parameters.

| Expert I.D. | Floor Area | Number of Floors | Slab Type | Internal Finishes | Elevator | External Finishes | Foundation Type | Basement | Electromechanical | Risk Management Application | Total Project Duration |
|---|---|---|---|---|---|---|---|---|---|---|---|
| Expert (1) | 4 | 4 | 1 | 3 | 1 | 3 | 1 | 2 | 2 | 3 | 4 |
| Expert (2) | 4 | 4 | 2 | 2 | 1 | 2 | 2 | 3 | 3 | 3 | 5 |
| Expert (3) | 3 | 4 | 1 | 1 | 1 | 1 | 1 | 2 | 3 | 4 | 4 |
| Expert (4) | 4 | 5 | 1 | 2 | 1 | 2 | 2 | 3 | 4 | 3 | 5 |
| Expert (5) | 3 | 4 | 2 | 2 | 2 | 2 | 2 | 2 | 2 | 3 | 4 |
| RII | 3.60 | 4.20 | 1.40 | 2.00 | 1.20 | 2.00 | 1.60 | 2.40 | 2.80 | 3.20 | 4.40 |
| Grade | H | VH | VL | L | VL | L | VL | L | M | M | VH |

The five experts agreed that the total duration of the project and the number of floors are the most important factors, with relative importance indices of 4.4 and 4.2, respectively, followed by the floor area, which has a relative importance index of 3.6. Experts agreed that the risk management application and the type of electromechanical factors are considered to have a medium effect on the cost estimation. While the factors of interior finishes, exterior finishes, and the number of basements were considered to have a low impact on cost estimation by experts. Whereas slab type, elevator number, and foundation type had very low impacts on cost estimation. As a result, any factor with an RII of less than 1.8 was eliminated from the final list. As a result, the slab type, elevator number, and foundation type were left off the final list of parameters influencing the total risk prediction in the conceptual stages. The final set of criteria consisted of the remaining eight parameters. Experts were asked to assess whether or not they agreed with the finalist list in the third round. Regarding the final list of criteria, which includes the remaining eight criteria, the experts agreed unanimously.

## 5. Data Collection

There were eight input parameters and one output variable in the data collected. Less than 200 square meters, 200 to 400 square meters, 400 to 600 square meters, and more than 600 square meters were the four categories for the floor space factor. The authors divided the factors of the number of floors into four categories: one or two floors, three to five stories, six to eight stories, and more than eight stories. The interior finishes variant is categorized into four groups: no interior finishes, basic, semi-finished interior finishes, and luxurious interior finishes. The choice is of the type of semi-finished interior finishes in the case of normal plaster for walls only and there are no paintworks, whereas for the type of basic interior finishes, it is in the case of the presence of paint works for the walls and ceramics for the floors. The type of luxurious interior finishes is chosen in the case of the presence of paint works for the walls and porcelain or marble works for the floors. The external finishing aspect was simply divided into two categories: basic and luxurious. The type of basic external finishing is chosen if the facades of the building have been painted only without any works of marble, Hashemite, or Pharaonic stone, while the external finishing is considered the luxurious type if the facades of the building have been done with any works of marble, Hashemite, or Pharaonic stone. There were two groups for the number of basements parameter: no basement and one basement. The overall project duration parameter was divided into four categories: less than six months, six months to a year, one year to two years, and more than two years. The risk management process application parameter was split into two categories: no risk management processes were performed on the project and risk management procedures were performed on the project. Electromechanical can be divided into two categories: basic and luxurious. The type of the electromechanical parameter is considered with the basic standards if the scope of work includes the main works of water, electricity, and sewage outside the apartment, but if it includes the internal works of the apartment, the type of the electromechanical parameter is considered a luxury type.

Based on the review of planned cost and actual cost data and using Equation (2), the authors assessed the overall percentage of risk for the completed projects. A project is excluded from the analysis if there is insufficient information about its planned cost or its actual cost. According to Table 5, the overall percentage of risk which is the major outcome variable was divided into three levels: low, medium, and high-risk scores.

$$\%OR = \frac{|AC - PC|}{PC} \times 100 \tag{2}$$

"*%OR*" represents the overall percentage of risk, "*PC*" represents planned cost and "*AC*" represents an actual cost.

**Table 5.** The Classifications of outputs.

| Category | Low | Medium | High |
|---|---|---|---|
| Impact on cost | Less than 10% | 10–20% | More than 20% |

The authors examined 250 projects and discovered that some data for the eight input variables or the result variable were missing. As a result, only the full data of 149 actual residential projects were accessed. For example, out of the 149 projects analyzed, "case no. the 26" project consisted of a 12-story building with 500 square meters per level, exquisite interior and exterior finishes, and luxurious electromechanical work. This building has one basement and was built in 20 months using risk management procedures, with an overall risk of roughly 12%. Due to the enormous population, it was assumed that the population's size was unlimited, so the sample size could be calculated using Equation (3). Table 6 shows demographic information about the respondents, whereas Table 7 shows demographic information about the inputs gathered from 149 projects.

$$SS = \frac{Z^2 \times p \times (1 - p)}{C^2} \tag{3}$$

where *SS* stands for sample size, *Z* stands for 1.96 with a 95% confidence level, *p* stands for the probability of selection, and *C* stands for the confidence interval. The sample size in this study was 149 projects, and the *p*-value was 0.5, hence the confidence interval was 0.08.

**Table 6.** The demographic data regarding the respondents.

| | | Work Experience in the Construction Industry | | |
|---|---|---|---|---|
| | | From 5 to 10 Years | From 10 to 15 Years | More Than 15 Years |
| Job title | Site engineer | 42 | 29 | 0 |
| | Project manager | 7 | 52 | 13 |
| | senior manager | 0 | 0 | 6 |

**Table 7.** Demographic data of inputs.

| Floor Area (A) | | Number of Floors (N) | | Internal Finishes (IF) | | Total Project Duration (D) | |
|---|---|---|---|---|---|---|---|
| Less than 200 | 28 | one or two | 36 | N.A | 20 | Up to 6 | 32 |
| 200 to 400 | 45 | three to five | 41 | half-finished | 49 | 6 to 12 | 40 |
| 400 to 600 | 46 | six to eight | 36 | Basic | 48 | 12 to 24 | 40 |
| more than 600 | 30 | more than 8 | 36 | luxury | 32 | more than 24 | 37 |
| **External finishes (EF)** | | **Number of basements (B)** | | **Risk management processes (RM)** | | **Type of electromechanical (E)** | |
| Basic | 72 | No | 118 | No | 132 | Basic | 77 |
| luxury | 77 | One | 31 | Yes | 17 | luxury | 72 |

Using Equation (4), the authors estimated Cronbach's Alpha for items. Cronbach's Alpha has a threshold of 0.7 [33]. Cronbach's alpha in this study was 0.757, which is higher than 0.7. It indicates that the scale is consistent and does not contradict itself, implying that it will produce the same findings when applied to the same sample again. Validity refers to how accurate a measurement is. The validity of this study was 0.87.

$$\propto = \frac{n}{n-1} \times \left(1 - \frac{\sum_1^n V_i}{V_t}\right) \tag{4}$$

where "$n$" represents the number of items, $V_i$ represents the variance of item $i$, and $V_t$ represents the variance of the test score.

## 6. Model Specification

The model was simulated using artificial neural networks. Due to its ease of use, IBM SPSS software was chosen to construct the model. It has a simple user interface and can be quickly imported and exported from Excel. The model has eight input parameters and just one output. Floor space, number of floors, interior, and exterior finishes, number of basements, total project time, risk management process application, and electromechanical type are all inputs. The output, on the other hand, is the overall risk factor. To evaluate the model's performance, the acquired data were randomly divided into 5-fold cross-validation. The first fold has 29 cases, whereas the subsequent folds have 30. Four folds were utilized to train the network in each model, while the fifth fold was used to evaluate the model. One hidden layer or two hidden layers might be present in a model. As a result, there are two different sorts of hidden layer groups. The number of neurons in the model with one hidden layer can be three, four, or five. In models with two hidden layers, the number of neurons can be four in the first layer and three in the second, or five in the first layer and three in the second, or five in the first layer and four in the second. Thus, there are three different groups in terms of the number of neurons in each hidden layer. The hyperbolic tangent function or the sigmoid function was employed as an activation function for the hidden layers, and both were investigated. Equation (5) can be used to estimate the number of models that can be tested. Twelve Multilayer Perceptron models were identified and tested as a result. The examined models and their mean absolute errors (MAE) in each k-fold are shown in Table 8. Equation (6) can be used to calculate the mean absolute error [34].

$$N_m = N_l \times N_a \times N_g \tag{5}$$

$$MAE = \frac{\left(\sum_{i=1}^N (ER - RS)\right)}{N} \tag{6}$$

where "$N_m$" stands for the number of models, "$N_l$" for the number of hidden layers, "$N_a$" for the number of hidden layer activation functions, and "$N_g$" for the number of neuron groups. "$ER$" stands for the model's estimated risk, "$RS$" for the risk score, and "$N$" for the number of case studies.

The mean absolute error of any model is equal to the mean error in its k-fold. Hence, the proposed model should have the minimum percentage of MAE. In this study, the MAE was equal to 11.7%, as shown in Table 8. The proposed model consists of two hidden layers: five neurons in the first hidden layer, and three neurons in the second hidden layer. The activation function of the hidden layer was the Hyperbolic Tangent function in the proposed model. Figure 1 illustrates the structure of the proposed model. The real and estimated overall risks are presented in Table 9.

**Table 8.** Mean absolute error of the models.

| Model | H3-0 | H4-0 | H5-0 | H4-3 | H5-3 | H5-4 | S3-0 | S4-0 | S5-0 | S4-3 | S5-3 | S5-4 |
|---|---|---|---|---|---|---|---|---|---|---|---|---|
| No. of hidden layer | 1 | 1 | 1 | 2 | 2 | 2 | 1 | 1 | 1 | 2 | 2 | 2 |
| No. of neurons in the first layer | 3 | 4 | 5 | 4 | 5 | 5 | 3 | 4 | 5 | 4 | 5 | 5 |
| No. of neurons in the second layer | - | - | - | 3 | 3 | 4 | - | - | - | 3 | 3 | 4 |
| Activation Function | H | H | H | H | H | H | S | S | S | S | S | S |
| K-1 | 11.4% | 10.7% | 13.4% | 10.7% | 13.4% | 13.4% | 14.8% | 15.4% | 18.1% | 18.1% | 15.4% | 13.4% |
| K-2 | 10.7% | 10.7% | 13.4% | 13.4% | 9.4% | 10.7% | 12.8% | 9.4% | 14.1% | 13.4% | 14.1% | 10.7% |
| K-3 | 12.8% | 16.1% | 16.8% | 16.8% | 14.8% | 16.1% | 20.8% | 15.4% | 16.8% | 15.4% | 12.1% | 17.4% |
| K-4 | 16.1% | 15.4% | 16.1% | 16.8% | 10.1% | 14.8% | 19.5% | 12.8% | 16.8% | 22.1% | 18.1% | 22.8% |
| K-5 | 10.7% | 14.8% | 16.1% | 14.8% | 10.7% | 16.1% | 15.4% | 16.8% | 18.8% | 14.8% | 19.5% | 20.1% |
| MAE | 12.3% | 13.6% | 15.2% | 14.5% | 11.7% | 14.2% | 16.6% | 14.0% | 16.9% | 16.8% | 15.8% | 16.9% |

"H" stands for the Hidden Layers' Hyperbolic Tangent activation function and "S" stands for the Hidden Layers' Sigmoid activation function.

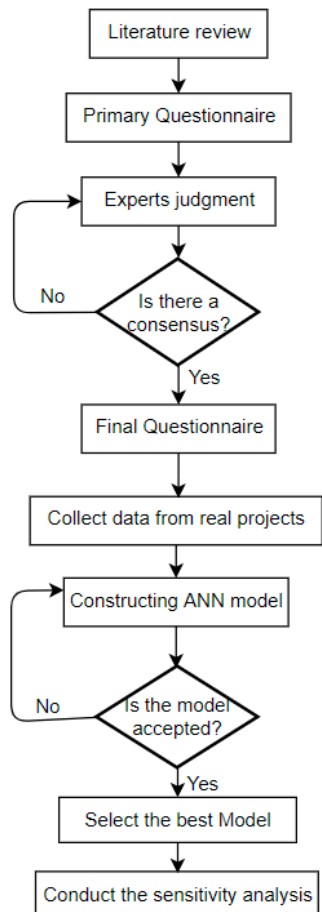

**Figure 1.** The Methodology of research.

**Table 9.** Classification of overall risk.

| Sample | Classification | Predicted | | | |
|---|---|---|---|---|---|
| | | **Low** | **Medium** | **High** | **Percent Correct** |
| Training | Low | 73 | 5 | 0 | 95.5% |
| | Medium | 5 | 26 | 1 | 85.1% |
| | High | 0 | 4 | 5 | 57.7% |
| Testing | Low | 14 | 2 | 0 | 92.6% |
| | Medium | 0 | 9 | 1 | 85.7% |
| | High | 0 | 4 | 0 | 23.1% |

## 7. Discussion

There is very little research on total risk assessment at the conceptual stage. For example, Oad et al. (2021) determined that price, the scope of work, and technical resources are the most important criteria in the bidding process at the conceptual stage [16]. Another study identified project cost, total project time, contract type, and use of risk management techniques as the primary criteria that can be used to assess overall risk in apartment buildings at the conceptual stage [15]. In this research, the main criteria used to estimate the total risks at the conceptual stage in residential projects are the number of floors, the building area, the interior finishes, the exterior finishes, the number of basements, the total duration of the project, the type of electromechanical, and the application of risk management processes. The scope of work that was identified as a critical factor in the study by Oad et al. (2021) was expressed in the current study by the number of floors, building area, internal and external finishes, and the number of basements. The duration of any activity can be estimated based on the required quantity and the production rate of the available resources. Hence, the technical resources identified by Oad et al. (2021) as critical factors in the conceptual stage were expressed in the quantities that can be inferred from the scope of work and the total duration of the project in the current study. Whereas the project cost component in the conceptual stage, which was identified by Badawy et al. (2022) can be estimated through many previous studies in the conceptual stage, which indicated that the cost can be deduced from the number of floors, building area, interior, and exterior finishes, and the number of basements, which was applied in the current study. Therefore, the eight input variables in the current study are in agreement with previous studies.

In the training phase, the proposed model predicted an average of 104 cases correctly and accurately with a ratio of 87.4% and predicted 15 cases incorrectly. The MAE for the low overall risk classification was 4.5%, and for the medium overall risk, the MAE was 14.9%. Unfortunately, the prediction of the overall risk in the case of the high-risk classification was 42.3%, which is considered a high ratio. In the testing phase, the proposed model predicted an average of 23 cases correctly and accurately with a ratio of 76.7% and predicted 7 cases incorrectly. The MAE for the low overall risk classification was 7.4%, and for the medium overall risk, the MAE was 14.3%. Unfortunately, the prediction of the overall risk in the case of the high-risk classification was 76.9%, which is considered a high ratio. Hence, the results indicated that this model is excellent in predicting the low and medium overall risk at the conceptual stage.

The mean absolute percentage error was 16.6% in an ANN model for estimating the total construction costs [26], while the MAPE was 13.2% in a hybrid technique for a cost assessment of residential projects at the early phase [21]. The MAPE was 26.3% an ANN approaches for cost forecast at the conceptual stage [32], while to estimate the cost of water treatment plants, the model has an error of 21.2% [29]. A model to predict the conceptual cost for engineering services in public construction projects was developed with a MAPE of 28.2 [30]. As a result of reviewing past research on conceptual-stage cost models, it was discovered that a mean absolute error of more than 13% was permitted, implying that the accepted model should have an error of less than 13%. The suggested strategy correctly classified 149 projects with a mean absolute error of 11.7%. Hence, this model can be accepted. The suggested model's acceptability implies that the eight input factors can be

utilized to predict the overall risk of residential projects at a conceptual stage. The results of the study agreed with the viewpoint of the five experts who were interviewed to determine the most important criteria in the final list that can be used to predict the overall risk in the conceptual stage of residential construction projects. The most important of these factors was the number of floors, which represents 28.5%. The second top criterion was the interior finishes with 16.3 percent. The execution of the risk management process component ranked third, with 14.4 percent, while the floor area element came fourth, with 11.7 percent. The total project time was the fifth component that had a 10.8% impact on the overall risk forecasting in the conceptual stages, followed by the exterior finishes, which had a 10.2% impact. Finally, the electromechanical type had a weight of 6.2%, and the lowest parameter was the number of basements with a relevance of 1.7 percent. The importance of each component in determining the overall risk in the conceptual stages of residential projects is depicted in Figure 2.

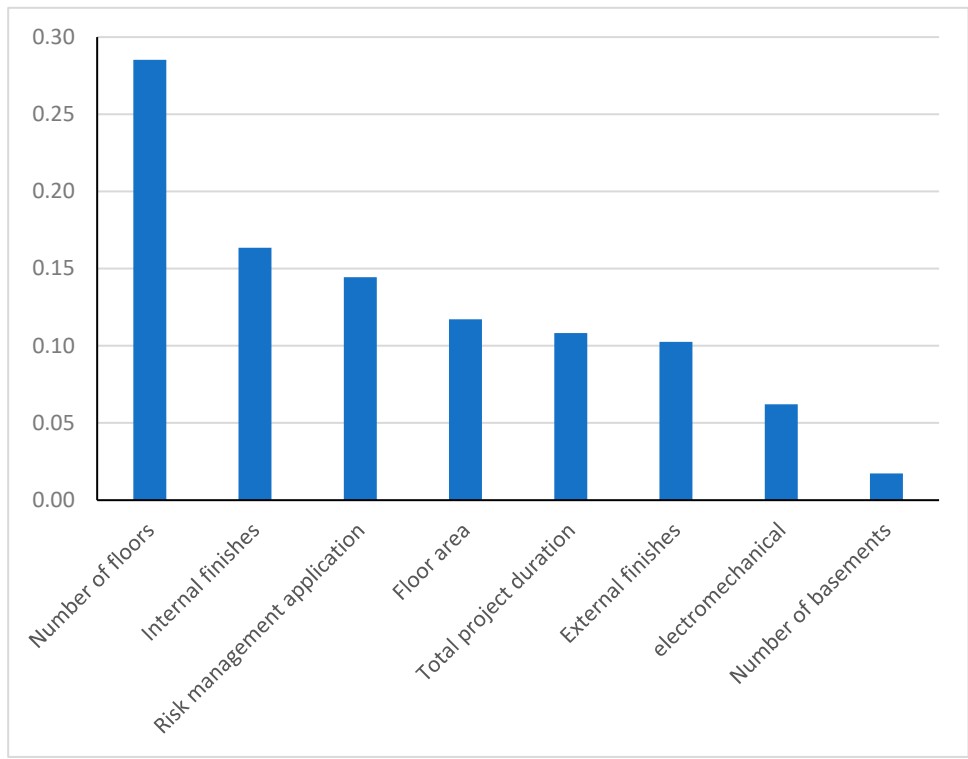

**Figure 2.** Importance of parameters in estimating the overall risk.

## 8. Conclusions

Decision-makers aim to predict the estimated value of the project budget in the conceptual stage to assess whether this investment is acceptable or not. The value of the reserve that covers the project's overall risk is included in the project budget. As a result, early on, a comprehensive risk assessment is required. There has been a great deal of research in developing cost estimation models, both in the planning phase and in the conceptual stages of a project. Unfortunately, there is a research vacuum in estimating risk in the conceptual stages of a project due to a lack of knowledge, so this study offers a model to forecast the overall risk at the conceptual stages of a project. A provisional list of essential characteristics, consisting of eleven parameters, was generated through a review of existing research and can be used to anticipate the overall risk in residential buildings at a conceptual stage. After three rounds of Delphi, the experts reached an agreement on the most critical parameters. The parameters for slab type, elevator number, and foundation type were omitted from the final list. Floor area, number of floors, interior finishes, external finishes, number of basements, kind of electromechanical, risk management process implementation, and overall project duration were all included in the final list. Four groups were created for

the floor space, the number of floors, interior finishes, and total project duration, while the internal finishes, the number of basements, the risk management method execution, and electromechanical kinds were all divided into two categories. Three levels were assigned to the output variable. Data were gathered from 149 actual residential projects. As a result, the confidence interval was 0.08 at the 95% confidence level. The model was simulated using artificial neural networks. The data were divided into five groups at random. There were twelve Multilayer Perceptron models identified and tested, each with a distinct number of hidden layers and activation functions. The proposed model has two hidden layers, the first of which has five neurons and the second of which has three neurons. In the suggested model, the Hyperbolic Tangent function was used to activate the hidden layer. The MAE was equal to 11.7% in this investigation. The number of floors is the most critical factor in determining the overall risk in the conceptual stages of residential projects, followed by interior finishing, and the risk management procedure. The electromechanical type and the number of basements were the least critical elements. The project manager can use the proposed model to identify residential projects in the conceptual stages as low-risk, medium-risk, or high-risk. As a result, the proposed model can assist stakeholders in deciding whether or not to continue with the project.

### 9. Limitations of Research

The overall risk and the influence of the important parameters were solely calculated based on the impact on the cost of the residential construction in this study. This study only looked at projects with fixed-price contracts. As a result, projects with cost-reimbursable contracts, for example, will require a re-estimation of the input parameter weights. The eight input criteria can be used in any country to obtain the overall risk at the conceptual phase. The data were obtained from 149 projects in Egypt, which means that the ranking of importance of each criterion may differ from one country to another. Hence, they should be double-checked the ranking of the importance of the criteria before being used in any other country. The user needs to alter the weights of the variables to adapt the model to subsequent times because the data used to produce it came from residential buildings in Egypt built between 2018 and 2020.

**Author Contributions:** Conceptualization, M.B. and F.K.A.; methodology, M.B.; software, M.B. and M.S.; validation, F.K.A.; formal analysis, M.S.; investigation, M.S.; resources, F.K.A.; data curation, M.B.; writing—original draft preparation, M.B.; writing—review and editing, F.K.A.; visualization, M.S.; supervision, M.B.; project administration, F.K.A.; funding acquisition, F.K.A. All authors have read and agreed to the published version of the manuscript."

**Funding:** This research was funded by the Research Supporting Project number (RSP-2021/264), King Saud University, Riyadh, Saudi Arabia.

**Informed Consent Statement:** Not applicable.

**Data Availability Statement:** The data collected and used for analysis will be available from the corresponding author upon request.

**Acknowledgments:** The authors extend their appreciation to the Research Supporting Project number (RSP-2021/264), King Saud University, Riyadh, Saudi Arabia for funding this work.

**Conflicts of Interest:** The authors declare that there are no conflicts of interest regarding the publication of this paper.

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
