# Peer review of "A Multilayer Perception for Estimating the Overall Risk of Residential Projects in the Conceptual Stage"

_buildings, doi:10.3390/buildings12040480_

Round 1

Reviewer 1 Report

Overall, this is a good paper that provides a new insight into risk in the conceptual stage. Nonetheless, there is some room for improvement.

  1. The authors need to include the aim and objectives of the paper in the introduction and further highlight the gap that this research fulfills.
  2. The concept of how the authors evaluated the overall percentage of risk for the completed projects has not been mentioned. Illustrating how the overall risk is calculated will help the readers better understand and utilize the model.
  3. The research is based in Egypt, yet the literature review shares very little information on how the residential projects in Egypt are different than other parts of the world and what are some of the risks that are exclusive to Egyptian residential projects.
  4. Lines 57 and 58 talk about the significant risks in construction projects in Egypt, Saudi Arabia. What about other parts of the world? It would be a good opportunity to provide the reader with some of the risks that occur worldwide or at least in the MENA region.
  5. Line 74, the author assumes that a conceptual stage overall risk estimation model’s acceptable rate is the same as that for cost-estimation models. Further elaborate on the basis that this assumption has been made.
  6. The methodology is very short, how was the Delphi method conducted? How were the experts chosen? A diagram would be a good way to articulate the author’s thoughts on how the methodology is conducted.
  7. The data collection section is informative and demonstrates that the sample size is sufficient. However, one comment would be that the measurements for some parameters can be expanded further. For instance, measuring the electromechanical parameter by basic and luxurious is vague. How can one identify what is luxurious and what is basic?
  8. The discussion part should explain how the results of this research compare to similar papers published in the field.
  9. In terms of language, some sentences need to be revised and improved. For example:
    1. Line 17 – 149 projects (add projects at the end of the sentence)
    2. Line 96 – Delphi technique (and not technology)
    3. Lines 120-122: Inconsistency: The foundation RII was less than 1.8, but it was kept as a parameter. While electromechanical components RII was 2.8 and it was removed. (According to table 3)
    4. Lines 149-150 – Fix the sentence (wrong grammatically)

Author Response

Reviewer 1:

The authors appreciate your valuable comments

Comment

Response

1.      The authors need to include the aim and objectives of the paper in the introduction and further highlight the gap that this research fulfills.

Thank you for this good comment

As there is a gap in developing a method to estimate the total risk in the conceptual stage, the aim of this research is to propose a model to predict the overall risk of residential buildings at the conceptual stage using an artificial neural network with a multilayer perception.

See the last paragraph in the introduction section from line 56 to line 59.

2.      The concept of how the authors evaluated the overall percentage of risk for the completed projects has not been mentioned. Illustrating how the overall risk is calculated will help the readers better understand and utilize the model.

I appreciate this excellent comment

Based on the review of planned cost and actual cost data and using equation (2), the authors assessed the overall percentage of risk for the completed projects. A project is excluded from the analysis if there is insufficient information about its planned cost or its actual cost. According to Table (5), the overall percentage of risk which is the major outcome variable was divided into three levels: low, medium, and high-risk scores.

                                                                     (2)

“%OR” represents the overall percentage of risk, “PC” represents planned cost and “AC” represents an actual cost.

See the second paragraph in the Data collection section from line 254 to line 261.

3.      The research is based in Egypt, yet the literature review shares very little information on how the residential projects in Egypt are different than other parts of the world and what are some of the risks that are exclusive to Egyptian residential projects.

I appreciate this good comment

The eight input criteria can be used in any country to obtain the overall risk at the conceptual phase. Whereas the data was obtained from 149 projects in Egypt, which means that the ranking of importance of each criterion may differ from one country to another. Hence, they should be double-checked the ranking of the importance of the criteria before being used in any other country.

See the Limitations of the research section from line 418 to line 422.

4.      Lines 57 and 58 talk about the significant risks in construction projects in Egypt, Saudi Arabia. What about other parts of the world? It would be a good opportunity to provide the reader with some of the risks that occur worldwide or at least in the MENA region.

Thank you for the significant comment;

There is a reference [8] on the risks in Egypt and Saudi Arabia

Reference [9] on risks in the United Arab Emirates

Reference [10] on the risks in the Kingdom of Saudi Arabia

Reference [11] on the risks in China

Reference [12] on hazards in New Jersey, United States

Reference [13] on risks in Kuwait

Reference [15] on risks in Egypt

Reference [16] on a systematic literature review for bid evaluation.

Furthermore, another reference [14] on risks in Nigeria has been added.

See the literature review section from line 61 to line 74.

5.      Line 74, the author assumes that a conceptual stage overall risk estimation model’s acceptable rate is the same as that for cost-estimation models. Further elaborate on the basis that this assumption has been made.

For every equation or model, there is a rate of error, but how can this ratio be judged, meaning how can determine if the model is accepted or not. It is not fair to judge the error rate of a model in the conceptual stage, where there is not enough information, to the error rate of a model in the design stage where there is sufficient information. It is expected that the error rate is less in the design stage than at the conceptual stage. Therefore, the error rate of any model must be compared with the extent of errors in the same stage. Since there are no researches that deal with estimating the total risk of the project in the conceptual stage, the error rate in the proposed model for calculating the total risk of the project in the conceptual stage was compared with the acceptable range of the rates of estimation models for cost estimation models in the conceptual phase.

See the literature review section from line 80 to line 90.

6.      The methodology is very short, how was the Delphi method conducted? How were the experts chosen? A diagram would be a good way to articulate the author’s thoughts on how the methodology is conducted.

Five experts with at least 15 years of experience in the construction of residential projects were randomly selected. The primary questionnaire was presented to the experts using the Delphi technique to determine the parameters in the final questionnaire in three stages. In the first stage, experts were asked to add any missing criteria, if any, that could affect the overall risk and could be discovered in the conceptual stage. Data is collected from experts, revised, and re-sent back to the experts where they are asked to rank each criterion on a five-point Likert scale. After collecting the data from the second round, the averages are calculated and any factor that has a very low impact on estimating the overall risk of the project is removed from the final list. In the third round, the experts are asked to assess whether or not they agree with the final list.

Figure (1) which illustrates the methodology of the research was added.

See the methodology section from line 160 to line 169 plus figure 1.

7.      The data collection section is informative and demonstrates that the sample size is sufficient. However, one comment would be that the measurements for some parameters can be expanded further. For instance, measuring the electromechanical parameter by basic and luxurious is vague. How can one identify what is luxurious and what is basic?

I appreciate this valuable comment

In future studies, it is possible to expand the division of some parameters.

In this research, the choice is of the type of semi-finished interior finishes in the case of normal plaster for walls only and there are no paintworks. Whereas for the type of basic interior finishes, it is in the case of the presence of paint works for the walls and ceramics for the floors. While the type of luxurious interior finishes is chosen in the case of the presence of paint works for the walls and porcelain or marble works for the floors.

The type of basic external finishing is chosen if the facades of the building have been painted only without any works of marble, Hashemite, or Pharaonic stone, while the external finishing is considered the luxurious type if the facades of the building have been done with any works of marble, Hashemite or Pharaonic stone.

The type of the electromechanical parameter is considered with the basic standards if the scope of work includes the main works of water, electricity, and sewage outside the apartment, but if it includes the internal works of the apartment, the type of the electromechanical parameter is considered a luxury type.

See the Data collection section from line 234 to line 253.

8.      The discussion part should explain how the results of this research compare to similar papers published in the field.

Although there is very little research on total risk assessment at the conceptual stage. For example, Oad et al. (2021) determined that price, the scope of work, and technical resources are the most important criteria in the bidding process at the conceptual stage [16]. Another study identified project cost, total project time, contract type, and use of risk management techniques as the primary criteria that can be used to assess overall risk in apartment buildings at the conceptual stage [15]. In this research, the main criteria used to estimate the total risks at the conceptual stage in residential projects are the number of floors, the building area, the interior finishes, the exterior finishes, the number of basements, the total duration of the project, the type of electromechanical, and the application of risk management processes. The scope of work that was identified as a critical factor in the study by Oad et al. (2021) was expressed in the current study by the number of floors, building area, internal and external finishes, and the number of basements. The duration of any activity can be estimated based on the required quantity and the production rate of the available resources. Hence, the technical resources identified by Oad et al. (2021) as critical factors in the conceptual stage were expressed in the quantities that can be inferred from the scope of work and the total duration of the project in the current study. Whereas the project cost component in the conceptual stage, which was identified by Badawy et al. (2022) can be estimated through many previous studies in the conceptual stage, which indicated that the cost can be deduced from the number of floors, building area, interior, and exterior finishes, and the number of basements, which was applied in The current study. Therefore, the eight input variables in the current study are in agreement with previous studies.

See the discussion section from line 328 to line 349.

9.      Line 17 – 149 projects (add projects at the end of the sentence)

Thank you for that good comment

Done

See line 17.

10.    Line 96 – Delphi technique (and not technology)

Thank you for that good comment

Done

See lines 162, 196.

11.    Lines 120-122: Inconsistency: The foundation RII was less than 1.8, but it was kept as a parameter. While electromechanical components RII was 2.8 and it was removed. (According to table 3)

Thank you for that good comment

A typo error and it has been corrected

As a result, the slab type, elevator number, and foundation type were left off the final list of parameters influencing the total risk prediction in the conceptual stages.

See the Identifying the critical parameters affecting the estimation of overall risk at the conceptual stages section from line 219.

12.    Lines 149-150 – Fix the sentence (wrong grammatically)

As a result, only the full data of 149 actual residential projects were accessed.

See the Data collection section from line 264 to line 265.

Reviewer 2 Report

In this paper is presented research dealing with the ability to foresee hazards that plays a critical role in estimating the entire project costs. It deals with a research vacuum in measuring the overall risk at this stage. Using artificial neural networks, proposed paper provides a strategy for estimating the overall risk in residential projects at the conceptual stage. There are eight important components in the suggested paradigm. The proposed model assists project managers in precisely estimating the project's overall risk, which leads to a more accurate estimation of the contract's entire worth at the conceptual stage, allowing the stakeholders to decide whether to proceed with the project.

Chapter 2. Literature review should be written with capital L not. Beside that this chapter can be improved – tables within this chapter (Table 1 and Table 2) should be explained in more details and especially references (16 to 23) within Table 1. Also it should be explained in more details how eleven criteria were defined because provided explanation is insufficient.

Chapter 3. Methodology is to short and needs to be extended in a manner that provides more insights about methodology. At least it is necessary to explain Delphi technology and an expert interview procedure at this point within paper.

The last sentence of this chapter: „The third step is to develop the model by experimenting with various hidden layer counts, numbers of neurons, and different hidden layer activation functions.“ is an insufficient to describe the modeling, training and testing of proposed ANN.

Chapter 4. Identifying the critical parameters affecting the estimation of overall risk at the conceptual stages should be extended by description of Table 4. The relative importance indices of critical parameters.

Author Response

Reviewer 2:

The authors appreciate your valuable comments

Chapter 2. Literature review should be written with capital L not. Beside that this chapter can be improved – tables within this chapter (Table 1) should be explained in more details and especially references (16 to 23) within Table 1.

To estimate the cost of school buildings in Korea, ten factors were identified. Three models were developed to calculate the cost of the school buildings, based on 217 projects. The first model was developed using neural network techniques, while the regression analysis was used in the second model and the third model was presented using the support vector machine. The results of the neural network model showed a more accurate estimate than the results of regression analysis or the supporting vector machine models [17]. Two studies were conducted in Gaza to estimate the cost of buildings at an early stage. The first research was based on seven variables and a model was proposed based on information derived from 71 construction projects using artificial neural networks [18]. While the second research developed a model for assessing the cost of construction projects with a high degree of accuracy and without the need for a lot of information, through the use of artificial neural networks. A database of 169 projects was collected from relevant institutions in the Gaza Strip has been adopted. The artificial neural network model has eleven factors as independent inputs [20]. A study to predict the cost of construction projects at the conceptual stage in Taiwan using ten parameters. The research suggested the utilization of the evolutionary fuzzy neural inference model to enhance cost assessment accuracy. The proposed model was relied on eleven factors [19]. In Egypt, a model to assess the cost of a residential building at an early stage using the artificial neural network and data obtained from 174 residential projects. The proposed model depended on four parameters: number of floors, the area of the floor, type of external finishing, and type of internal finishing [21]. The costs of 136 executed projects were utilized to propose an artificial neural network model to predict the preliminary cost of construction projects in Yemen. The suggested model contained 17 factors [22]. In United State, research was conducted on the difference in the computation of construction costs utilizing artificial neural networks by comparing nineteen variables in 20 projects [23]. In Taiwan, a study has presented a prototype for the rapid assessment of a proposal integrating a probabilistic cost sub-model and a multi-factor assessment sub-model. The cost-based sub-model concentrates on the cost divisions. While the multi-factor assessment sub-model captures the specific elements influencing the cost division. That research is based on 21 variables [24]. The eight previous studies almost agreed on nine primary factors that can be used for cost estimates at the conceptual stage of a project. These nine factors are floor area, number of floors, type of foundation, number of elevators, type of slab, type of exterior finishing, interior finishes, type of electromechanical works, and number of basements. Table 1 shows the different sources for each factor.

See the literature review section from line 92 to line 126.

Chapter 2. this chapter can be improved – tables within this chapter (Table 2) should be explained in more details.

There is a lot of research that investigates the cost estimates at the conceptual stage. Each research proposed a model with a mean absolute percentage error. Table 2 shows the mean absolute percentage error for some of these prior studies. Which illustrated that the errors in the proposed models were ranged from 4 – 28.2 %. This means that the maximum acceptable mean percentage error in the proposed model at the conceptual stage is 28.2%.

See the literature review section from line 136 to line 141.

Also it should be explained in more details how eleven criteria were defined because provided explanation is insufficient.

Limited studies in the conceptual stage of risk estimation identified four criteria: use of risk management processes, duration of the entire project, total cost, and type of contract. As the total cost of the project can be estimated through nine criteria: floor area, number of floors, type of foundation, number of elevators, type of slab, type of external finishing, internal finishes, type of electromechanical works, and number of basements. Hence, the cost of the project can be replaced by these nine factors. While the type of contract was not included due to research limitations, as the research is related to estimating the cost of housing projects based on a fixed price contract only. Hence, the initial list of criteria used to derive the overall project risk at the conceptual stage contained eleven criteria: floor area, number of floors, type of foundation, number of elevators, type of slab, type of exterior finish, interior finishes, type of electromechanical works, number of basements plus the use of risk management processes, and the entire project duration.

See the literature review section from line 143 to line 154.

Chapter 3. Methodology is to short and needs to be extended in a manner that provides more insights about methodology. At least it is necessary to explain Delphi technology and an expert interview procedure at this point within paper.

Five experts with at least 15 years of experience in the construction of residential projects were randomly selected. The primary questionnaire was presented to the experts using the Delphi technique to determine the parameters in the final questionnaire in three stages. In the first stage, experts were asked to add any missing criteria, if any, that could affect the overall risk and could be discovered in the conceptual stage. Data is collected from experts, revised, and re-sent back to the experts where they are asked to rank each criterion on a five-point Likert scale. After collecting the data from the second round, the averages are calculated and any factor that has a very low impact on estimating the overall risk of the project is removed from the final list. In the third round, the experts are asked to assess whether or not they agree with the final list.

Figure (1) which illustrates the methodology of the research was added.

See the methodology section from line 160 to line 169 plus Figure 1.

The last sentence of this chapter: „The third step is to develop the model by experimenting with various hidden layer counts, numbers of neurons, and different hidden layer activation functions.“ is an insufficient to describe the modeling, training and testing of proposed ANN.

The third step is to develop the model. The model is simulated using artificial neural networks using IBM SPSS software. The critical parameters that affect the estimation of the overall risks at the conceptual stages are considered as inputs to the model while the output is the overall risk of the project. The model may contain one or two hidden layers. The number of neurons in the one-hidden-layer model can be three, four, or five. In models with two hidden layers, the number of neurons can be four in the first layer and three in the second, or five in the first layer and three in the second, or five in the first layer and four in the second. Thus, there are three different groups in terms of the number of neurons in each hidden layer. Hence, there are six models that can be developed. As the hyperbolic tangent function was used as an activation function for the hidden layers in six models, and the Sigmoid function was tested as an activation function for the hidden layers in six other models. Hence, twelve Multilayer Perceptron models have been identified and tested. To evaluate the performance of the model, the available data were also randomly divided into 5-folds. The first fold contains 29 cases, while the subsequent folds contain 30 cases. Four folds were used to train the network in each model, while the fifth fold was used to evaluate the model. The final proposed model for estimating the overall risk in the conceptual stages is the model with the lowest mean absolute error rate.

See the methodology section from line 169 to line 186.

Chapter 4. Identifying the critical parameters affecting the estimation of overall risk at the conceptual stages should be extended by description of Table 4. The relative importance indices of critical parameters.

The very low category has a range of 1 to 1.8. The low category has a range from 1.8 to 2.6, while the range of the medium category is from 2.6 to 3.4. The high category is from 3.4 to 4.2, whereas the very high category is from 4.3 to 5.0. The five experts agreed that the total duration of the project and the number of floors are the most important factors, with relative importance indices of 4.4 and 4.2, respectively, following by the floor area, which has a relative importance index of 3.6. Experts agreed that the risk management application and the type of electromechanical factors are considered to have a medium effect on the cost estimation. While the factors of interior finishes, exterior finishes, and the number of basements were considered to have a low impact on cost estimation by experts. Whereas slab type, elevator number, and foundation type had very low impacts on cost estimation.

See the Identifying the critical parameters affecting the estimation of overall risk at the conceptual stages section from line 207 to line 217.

Round 2

Reviewer 1 Report

The authors have addressed my concerns.